# Development of Low-Cost Portable Spectrometers for Detection of Wood Defects

**DOI:** 10.3390/s20020545

**Published:** 2020-01-19

**Authors:** Jakub Sandak, Anna Sandak, Andreas Zitek, Barbara Hintestoisser, Gianni Picchi

**Affiliations:** 1InnoRenew CoE, Livade 6, 6310 Izola, Slovenia; anna.sandak@innorenew.eu; 2Andrej Marušič Institute, University of Primorska, Titov trg 4, 6000 Koper, Slovenia; 3Faculty of Mathematics, Natural Sciences and Information Technologies, University of Primorska, Glagoljaška 8, 6000 Koper, Slovenia; 4FFoQSI—Austrian Competence Centre for Feed and Food Quality, Safety & Innovation, FFoQSI GmbH, Technopark 1C, 3430 Tulln, Austria; andreas.zitek@ffoqsi.at; 5Institute of Wood Technology and Renewable Materials, University of Natural Resources and Life Sciences, Konrad Lorenz-Straße 24, 3430 Tulln an der Donau, Austria; barbara.hinterstoisser@boku.ac.at; 6CNR-IBE, Via Madonna del piano 10, 50019 Sesto Fiorentino, Italy; gianni.picchi@cnr.it

**Keywords:** NIR spectroscopy, wood defects, portable instruments, in-filed measurement

## Abstract

Portable spectroscopic instruments are an interesting alternative for in-field and on-line measurements. However, the practical implementation of visible-near infrared (VIS-NIR) portable sensors in the forest sector is challenging due to operation in harsh environmental conditions and natural variability of wood itself. The objective of this work was to use spectroscopic methods as an alternative to visual grading of wood quality. Three portable spectrometers covering visible and near infrared range were used for the detection of selected naturally occurring wood defects, such as knots, decay, resin pockets and reaction wood. Measurements were performed on wooden discs collected during the harvesting process, without any conditioning or sample preparation. Two prototype instruments were developed by integrating commercially available micro-electro-mechanical systems with for-purpose selected lenses and light source. The prototype modules of spectrometers were driven by an Arduino controller. Data were transferred to the PC by USB serial port. Performance of all tested instruments was confronted by two discriminant methods. The best performing was the microNIR instrument, even though the performance of custom prototypes was also satisfactory. This work was an essential part of practical implementation of VIS-NIR spectroscopy for automatic grading of logs directly in the forest. Prototype low-cost spectrometers described here formed the basis for development of a prototype hyperspectral imaging solution tested during harvesting of trees within the frame of a practical demonstration in mountain forests.

## 1. Introduction

Wood log grading is an action used to determine a set of characteristics regarding wood quality that are later used by forest resources managers, traders and manufacturers. Overall log quality and prospective end use have a major influence on the log’s economic value. The grader’s challenge is to properly assess all visible characteristics of each log, including the log’s geometry and presence/location of defects. Subsequently, the expert can identify the quality class corresponding to such a unique set of characteristics. Strict following of clearly defined assessment procedures as well as an objective grading verdict corresponding to established grading rules must be guaranteed along the whole inspection process [1]. Consequently, the role of the expert grader is to assure proper assessment and, particularly, determine the log’s gross dimensions to estimate what portion of the log is available to produce a given product and evaluate the quality of the product that could be produced from the log.

The procedure for log quality grading is formalized and described in dedicated standards (e.g., EN 1309, EN 1316) or established as commonly recognized procedures [2]. Grading results in a quality class, typically A, B, C or D, with the first considered as superior. In that case, a trained expert visually detects all wood imperfections or defects as well as manually measures log dimensions and geometrical features. Commercial value of the resource is determined after combining all the available information into the unique quality grade. An important limitation of this approach is that it relies on visual assessment by a person, which, from its nature, is subjective and of limited reliability/repeatability. Industries, therefore, wish to adopt more objective methodologies for resource quality determination by integrating sensors and novel descriptors of the quality, better adjusted to specific needs of the value chain players [3].

Wood defects are abnormalities or irregularities found in wood that may be responsible for reducing wood strength, durability, appearance and, as a consequence, its economic value. Defects investigated in this work were related to the natural growth of trees, which has an effect on development of wooden tissue during tree lifetime under certain environmental conditions and influences. Recently used methods rely on visual rating, which is subjective, operator-dependent, time-consuming and not precise. Therefore, it is desired that grading should be conducted by means of automatic assessment to assure a faster, more reproducible and reliable way of quality sorting.

Near infrared (NIR) spectroscopy has great potential for wood quality evaluation [4,5], mapping of wood properties [6,7] and mechanical properties estimation [8]. However, its practical application in field/forest is challenging [9,10,11]. Portable NIR instruments were successfully used in the field for tree breeding [12,13,14], prediction of tracheid length [15], assessment of wood and fiber properties in standing mountain pine beetle-attacked trees [16], wood species recognition [17], wood moisture content prediction [18] and estimation of leaf quality [19]. However, until now, only a few applications have been implemented in real-world wood processing industries, mostly for on-line sorting of wooden products and quality control of production [20,21,22].

Vibrational spectroscopy techniques are well suited to be used as portable or handheld due to their speed, selectivity, simplicity and no need for samples preparation [23]. Recent hardware development, combined with a miniaturizing trend in highly performing sensors and accessories, has enabled the development of state-of-the-art portable NIR equipment dedicated for forest/wood industries. Such instruments should operate in harsh environments with due precision, reliability and accuracy. Relatively fast measurement time, lightweight and ergonomic design, intuitive user interface and absence of moving parts makes such equipment an interesting alternative for in-field and on-line measurements [24,25,26].

The overall objective of this work was to use spectroscopic methods as an alternative to visual grading usually performed while assessing wood quality. Sensors were intended both for manual use and installation on timber processor heads, making possible an automatic grading of logs during their production in the forest or at the roadside. For that reason, an original solution for a low-cost and portable spectroscopic instrument capable of automatic detection of selected wood defects was built and examined.

## 2. Materials and Methods

### 2.1. Sample Collection

Twenty-five wood discs of Norway spruce (*Picea abies* L. Karst) cut from trees of diameters between 100 and 400 mm were collected in Forchtenstein (Burgenland, Austria) in the spring of 2015. Directly after sampling, all samples were wrapped in aluminum and plastic foil, frozen and stored at −21 °C in order to avoid wood drying and/or biological degradation. Laboratory measurements of the defrosted samples were performed at a surface temperature of approximately 15 °C, without any additional sample conditioning or surface preparation. The target set of researched logs contained several wood defects frequently present in naturally grown trees, such as resin pockets, compression wood, knots and decay. Some examples of the experimental samples and diverse wood deficiencies identified on the disks’ surface are shown in Figure 1. In addition to defected samples, a set of optimal quality wooden disks (containing normal wood tissue and bark) was collected in order to allow direct comparison between diverse quality classes. As a result, it was possible to measure spectra of at least 36 independent replicas for each defect.

### 2.2. Spectroscopic Measurements

Three portable spectrometers covering different spectral ranges were selected for testing. A summary of technical characteristics is presented in Table 1 and Figure 2.

The commercially available sensor MicroNIR Pro 1700 produced by VIAVI (Santa Barbara, CA, USA) was used for both laboratory and in-field measurements. The sensor is compact and includes all components (linear variable filter, CCD detector, light source and processing unit) integrated within a single easy-to-handle assembly. The sensor was connected to a portable laptop computer by USB cable and did not require any additional electric power supply. Spectra were acquired with a scanning frequency of 80 Hz allowing 12.5 ms of the integration time. The effective spectral range covered by the senor was 950–1650 nm (10,526–6060 cm^−1^), with the extreme parts of the spectra excluded due to the low signal-to-noise ratio. A custom software for data acquisition and post-processing was developed in LabView by integrating tools available in the Software Development Kit.

In addition, Hamamatsu C12880MA and C11708MA (Hamamatsu City, Japan) micro-spectrometers were identified as low-cost instruments potentially useful for measuring experimental samples. Both sensors cover different spectral ranges, including 340–780 nm (29,411–12,820 cm^−1^) and 640–1050 nm (15,625–9524 cm^−1^) for HamamatsuVIS (C12880MA) and HamamatsuNIR (C11708MA), respectively. The commercially available hardware included only a spectrometer module integrating optical and electronic components assembled together in a compact and resistant box. It was necessary, therefore, to build fully functional instruments by adding focusing optics, illumination and signal processing units. Detailed description of the developed prototype instruments is presented in the following chapter.

Portable instruments were compared with a benchmark FT-NIR MPA (Fourier Transform-Near infrared Multi-Purpose Analyzer) spectrometer produced by Bruker Optics (Ettlingen, Germany). Absorbance spectra were acquired by means of a fiber optic probe covering the measurement area of 10 mm^2^. The spectral range of the FT-NIR instrument was 1000–2380 nm (10,000–4200 cm^−1^) and the zerofilling factor was 2. Fifty scans per spectrum were collected with a resolution of 8 cm^−1^. Spectra acquired by FT-NIR instrument were averaged to minimize natural variability due to anatomical alterations within investigated wood samples. Therefore, these were used as model spectra related to specific wood defects in order to highlight physical-chemical differences as recorded by near infrared.

### 2.3. Prototypes Development

The core of the first prototype sensor was micro-spectrometer Hamamatsu C11708MA, capable of measuring electromagnetic radiation in the near infrared spectral range with a spectral resolution of 20 nm. The sensor itself is compact (thumb size: 27.6 × 16.8 × 13 mm^3^) and light (weight: 9 g). However, it does not include any focusing optics nor built-in illumination. For that reason, one × 60 Zoom Mini Phone Camera Lens Microscope Magnifier was installed in front of the spectrometer and a second lens in front of the illuminating bulb, as shown in Figure 3. The solution was cost-effective and allowed measurements at an optimal distance of D = 3 mm between lenses and the object surface.

Selection of the optimal illumination was the most challenging task and several light sources (including LED illuminators, halogen and fluorescence bulbs) were chosen for testing during the preparatory phase. Images of some evaluated bulbs as well as responses from the HamamatsuNIR spectrometer are presented in Figure 4. In this case, the spectra of the light reflected from the reference surface (Spectralon) are plotted together. Considering its compact size, sufficient light intensity and optimal coverage of the complete spectral band, bulb #3 was selected for integration within the prototype instrument. All parts, including spectrometer, optics, illumination and controller unit, were assembled together in a simple carton box protecting the optical system from external illuminations (Figure 3).

A similar procedure was implemented for setting up the second sensor (Hamamatsu C12880MA), where identical light source (bulb #3) and focusing optics were integrated with the spectrometer.

The electronic system used for controlling both spectrometers was developed on Arduino UNO microcontroller frames. The simplified wiring schema for connections is presented in Figure 5. The system worked properly in that configuration, even though it is recommended to add digital buffers for low power electronic circuits. The basic software code uploaded to the microcontroller internal memory is provided as Appendix A and Appendix B for Hamamatsu C11708MA and C12880MA, respectively.

Total cost of the hardware required for implementing the prototype in-field systems presented here varied from a few hundred (Hamamatsu sensors) to several thousands (MicroNIR sensor). However, additional resources should be considered when integrating these with a dedicated controller (instead of a laptop PC as used in this study) as well as all-weather cover and protection. Even then, overall investment to that solution is far less costly when compared to the corresponding laboratory equipment (Fourier transform infrared (FT-NIR)) and still provides a unique set of quality characteristics impossible to acquire with alternative technologies.

### 2.4. Data Evaluation and Mining

Two alternative classification methods, PLS-DA (Partial Least Square Discriminant Analysis) and SVMC (Support Vector Machine Classifier) were used for spectroscopic data post-processing. PLS_Toolbox 8.0 (Eigenvector Research, Manson, WA, USA) and LabView 13 (National Instruments, Austin, TX, USA) were used as software platforms for spectral data analysis and the generation of chemometric models. The best configuration for the model was identified with the Model Optimizer software tool as part of the PLS_Toolbox package. Optimization included testing of all combinations for discrimination algorithms, principal components number, spectra pre-processing and selected ranges of spectra. Extended Multiplicative Scatter Correction (EMSC), spectra smoothing, first and second derivatives (Savitzky-Golay), as well as area normalization, were used as spectra pre-processing routines.

## 3. Results and Discussion

### 3.1. Quality Parameters of Logs Detectable by NIR Spectroscopy

Visual grading is the traditional method for determining log quality and is based on types, sizes and positions of physical characteristics that are not allowed for each quality class. A short description of wood defects downgrading Norway spruce log quality class is briefly summarized below.

Knots are the portion of a branch or limb that has been surrounded by subsequent xylem grown during tree life. The size, type and distribution of knots have the most important impact on lumber mechanical resistance and are main considerations when applying grading rules. Knot size (usually diameter) is measured in centimeters. The severity of grain deflection caused by the knot is correlated with its size. The knot reduces lumber (log) quality when its diameter is >2 cm. The distribution of knots in logs depends on the tree species and position within the stem. It is also determined by the growth characteristics of the stand and the age of the tree among the others.

Resin pockets are small discontinuities within the xylem that are filled with resin and wound tissue. They are usually occluded after some years of wood formation and tree growth. The occurrence of resin pockets is characteristic for softwoods possessing resin canals (such as Norway spruce) or trees exposed to stresses (animal/insect attacks, sites exposed to winds, etc.). The influence of resin pockets on the strength properties is insignificant; however, it downgrades visual appeal of products made from such wood and may result in continuous release of the resin on the wood surface when it is used for high-end finishing products (e.g., clear wood, windows, veneers).

Compression wood is a type of defect that tends to form in conifers exposed to strong winds or trees growing on a slope. Compression wood is often characterized by a dense hard brittle grain and reacts to seasonal moisture changes. Properties of compression wood are considerably different from those in normal mature wood. Compression wood tracheids, for example, are about 30% shorter than normal and have higher microfibril angle. In addition, compression wood contains about 10% less cellulose and 8–9% more lignin and hemicelluloses than normal wood. These factors reduce the desirability of compression wood for pulp and paper manufacture. Compression wood not only yields less cellulose but produces low strength pulp. The general effects of compression wood on the performance of sawn timber are reduction in the strength, stiffness and dimensional stability, resulting in a decrease in yield of high-quality end products. Compression wood may cause problems in processing the log by exhibiting bow and spring in the manufactured product.

Rot is caused by a variety of fungi that break down and digest wood chemical components. Fungi attack wood according to their specific enzymatic system. Brown-rot fungi have limited impact to lignin structure; however, they easily degrade polysaccharides. White-rot fungi have the capability to degrade mainly lignin structure but also carbohydrate wood cell components (cellulose and hemicellulose) [27]. Rot is the wood defect for which quality grade reduction is obligatory. Rot fungus enters the tree through a root, broken branch, damaged treetop or scar on the stem. Logs cut from older trees are suspected to contain more developed rot. The final stage of rot is a complete or extensive material loss forming internal cavities.

Finally, bark, which is not a wood defect but the tree’s external protective layer, might be detected by spectroscopy. Bark tissues make up 10–20% of tree weight and consist of various biopolymers, tannins, lignin, suberin, suberan and polysaccharides [28]. The wood/bark ratio is an important parameter that affects lumber yield [29].

### 3.2. Model Spectra of Wood and Wood Defects

NIR spectrometers are highly suitable for assessment of heterogeneous organic matter, including living trees and wood [3,10,30]. The NIR spectrum contains information regarding both chemical composition and physical state of measured samples. The spectral peak position and its shape corresponds to the presence of specific functional groups possessing dipole momentum. The low frequency component of spectra (scatter) is related to the optical properties of the matter and highly depends on sample preparation and presentation. FT-NIR spectroscopy, as the most sensitive technique among those tested in this research, was selected to highlight effects of wood defects on the spectral fingerprint. Therefore, Figure 6 presents a second derivative spectra for all wood deficiencies investigated. The interpretation of spectra includes identification of spectral features differentiating its outline and assigning these according to the literature references [31,32] as summarized in Table 2.

An unwanted effect of the derivatization process is that the signal-to-noise ratio decreases at higher orders of derivatives. It is a consequence of the discrimination effect and the fact that, by its nature, the noise contains the sharpest features in the spectrum. As a consequence, the bandwidth of noise corresponds to the interval of the spectral data used in the derivative calculation. The decrease in signal-to-noise ratio can be reduced by using smoothing properties of the Savitzky-Golay algorithm, even if a high degree of smoothing can distort the derivative spectrum [33].

The most dissimilar spectrum was that corresponding to the resin pocket. Resin serves to protect the tree from dehydration and microbial attack and is typically composed of monoterpenes and resin acids [34]. Resin pockets spectra contain unique spectral bands not present in other wood features (5700, 5812, 5909, 6117, 7344 cm^−1^). All of the above are assigned to -CH or -CH_2_ groups of aromatic and aliphatic chains in wood extractives [32]. Knots, compression wood and bark contain more lignin; therefore, their band at 5980 cm^−1^ is more profound as compared to the reference spectrum of normal wood. The model spectrum of decayed wood appeared to have similar shape to that of normal wood. However, it was noticed that the decay on the samples investigated was at the very early stage when degradation of woody polymers was still negligible. Therefore, it was problematic to assuredly assign specific decay type (white or brown rot) on the basis of FT-NIR spectra only.

### 3.3. Recognition of Wood Defects by Automatic Classification Methods

Two alternative chemometric methods, PLS-DA and non-linear SVMC, were used for spectral data classification. Both methods belong to the group of supervised techniques and require complete information regarding the membership of each sample/measurement to a certain category. The algorithm is capable (after proper calibration) to classify a new/unknown sample into one of the pre-defined classes on the basis of its spectral pattern [35].

PLS-DA is a multivariate inverse least-squares analysis method used to classify samples. It decomposes the spectra as linear combinations of principal components (PC) that express the major part of information contained in the overall dataset. The predictor variables or latent variables (LVs) are generated from the input variables to maximize the variance between sample classes in the model [36]. PLS-DA is widely used in the analysis of multivariate data. It helps to determine if groups of samples are distinct and identify all the spectral features that can describe the differences between groups. All these differences are expressed as model loadings or latent variables. An important advantage of PLS-DA is its availability to handle highly colinear and noisy data. Moreover, it provides a visual interpretation of a complex dataset through easily interpretable scores plots that highlight the differences between groups.

SVM is a flexible method that makes no assumption regarding data. It is a nonlinear classification method that constructs a set of hyperplanes in a high or infinite dimensional space and a good separation is achieved by the hyperplane that has the largest distance to the nearest training data point of any class [36]. It works by obtaining the optimal boundary of two groups in a vector space independent of the probabilistic arrangements of vectors in the training set. When the linear boundary in low dimension input space is not enough to separate two classes, SVM can create a hyperplane that allows linear separation in the higher dimension feature space [35]. Support Vector Machine is a supervised machine learning algorithm useful for solving both classification and regression problems. In this approach, data are presented in *n*-dimensional space, corresponding to the number of variables describing the data. In contrast to PLS-DA, SVM is not influenced by the distribution of diverse sample classes. The methods allow flexibility in the choice of kernel function that leads to the separation of two groups of samples by solving either linear or non-linear problems. An important limitation is that SVM does not provide interpretable model’s statistics, such as scores or loadings [37].

Results of the PLS-DA classification for MicroNIR and both prototype Hamamatsu sensors are presented in Figure 7 in a form of the confusion table. All columns located on the diagonal running from bottom left to top right (excluding unsigned row) correspond to the properly identified samples. In that case, the spectra of a given defect was identified as a member of the class describing exactly such defect. Therefore, it is desired that the optimal discrimination model results in majority of validation samples appearing on the diagonal of the confusion table. All the other spectra (not laying on the table diagonal) are considered as undesirable or wrong classifications. The case when normal wood is identified as defect is defined as false negative, in contrast to false positive when deficient wood as identified as normal. A convenient numerical quantifier of the discrimination model accuracy is the success rate (*SR*) computed as a percent ratio of the correctly classified to total number of samples. It has to be noted that more than 1800 independent spectra were used for the models development, considering 66% of spectra used for calibration and 34% for independent data set validation.

The PLS-DA model capably fit the microNIR spectral data and only five lateral variables (explaining 95.4% of the variance) were enough to reduce the cross-validation error of prediction below 15%. The outline of loadings (variables) is presented in Figure 8, where all the spectral features allowing proper samples differentiation can be identified. The classification decision was taken by the system considering two alternative approaches. The first algorithm identifies the class considering the highest value of the affinity probability among the all confronted classes. In that case, all samples are assigned to only one class and none belongs to the “unsigned.” The second algorithm confronts the PLS model results with a predefined threshold. The sample is assigned to a given class if it passes the threshold; otherwise, it is considered as “unsigned.” The second algorithm is more reliable but also very sensitive to minor disturbances within the source spectra. For that reason, in case of real-word implementation of the discrimination of wood defects, the “Prediction Most Probable” approach is recommended. It is clear that the best performance of wood defects discrimination was by modeling spectra acquired with microNIR spectrometer (success rate *SR* = 91%). However, low-cost prototype sensors were also capable of properly identifying wood defects, even though the success rates were lower (58% and 74% for Hamamatsu VIS and NIR, respectively). The most problematic was the differentiation between knots and reaction wood. This was due to several physical-chemical similarities between both defects and also limited spatial resolution/accuracy of prototype spectrometers. The high number of unsigned samples identified with “Prediction Strict” algorithm highlights the drawback of this method and confirms the high similarities between spectra obtained on the wood surface.

The greatest advantage of PLS-DA is a possibility for the model interpretation by means of loadings. However, the practical implementation may not be optimal as not all the variance within source data may be represented by a limited number of principal components. Therefore, it is frequently applied to use machine learning algorithms for data representation and identification. Support Vector Machine is one of these routinely used tools for spectral data processing and was, therefore, implemented to assess the identical data set as in PLS-DA. Confusion tables for resulting SVMC models obtained are presented in Figure 9, assuming both classification rules. The success rate for all sensors is evidently higher than those obtained with PLS-DA. Only a few samples were misclassified, mostly in the case of Hamamatsu NIR and resin mixed with normal wood. The performance of the microNIR sensor was nearly faultless even if implementing “Prediction Strict” rule for classification. The number of unsigned samples was considerably lowered for both Hamamatsu prototype sensors. It was assumed, therefore, that the overall performance of the SVMC was superior to PLS-DA. Unfortunately, no direct interpretation of the model was possible and only the independent validation was proof of the model reliability.

## 4. Conclusions and Practical Recommendations Regarding the Implementation of the (Prototype) Sensors for Wood Defects Detection

Portable spectroscopic instruments are an interesting alternative to laboratory equipment, allowing fast and reliable measurements for chemical/physical properties of materials. In this work, one commercially available and two self-constructed prototype instruments were tested with the purpose of wood defects discrimination. The performance of all spectrometers was compared by means of PLS-DA and SVMC methods.

The most convenient place to measure spectra with sensors developed in this research is on the cross section of logs after tree felling or on the pile of stored logs. Utilization of spectrometers to asses log sides is problematic as the bark layer prevents light from accessing the log’s interior. The only information available to scrutinize are traces of knots appearing on the side of the log visible over the bark layer. Even if determination of the log’s quality by assessing its cross section may be considered as limited, it can provide an enormous possibility to improve the state-of-the-art in quality grading and managing of value chains. The functionality of the proposed low-cost spectroscopic system can be extended to predict chemical composition of wood polymers as well as its distribution (map) over the log section or whole volume. It can lead toward development of new methodologies where raw resources are optimally destinated to down-stream conversion following the paradigm of “proper wood resources for proper use”.

The up-to-date results and experiences highlight the MicroNIR sensor as a superior technique for fast and automatic discrimination of wood deficiencies. It is especially advantageous as it is a compact system integrating all optical and electronic components. It allows minimization of electrical noises and, therefore, increases overall signal-to-noise ratio. An important drawback of this solution is its relatively high cost, especially when considering the use of an array of sensors for better spatially defined analysis in-field or on-line. The prototype low-cost sensors, therefore, may be a good alternative, allowing further customization of the technical solution and, consequently, improvement of the system’s performance. Both, hardware assembly and necessary software, are rather simple for implementation, offering a wide range of possibilities for the sensor’s fit-to-purpose design.

An important issue is proper spatial calibration of the hardware. The field of view is controlled in our solution by adjusting the focusing optics and distance between the lenses and measured surface. However, the depth of field for microscopic lenses is very narrow, resulting in likely blurring of the image and reduction of light quality reaching the spectrometer. It is especially significant when evaluating wooden disks having excessive surface roughness resulting from log cutting with a chain saw.

As a rule of thumb, it is necessary to assure sufficiently frequent white and black reference calibration of the sensor while working with portable equipment. This is due to the fact that resulting spectra is highly affected by the sensors and surface temperatures recurrently changing when operating in-field. Previous studies [38,39] reported that temperature variations degrade the quality of agri-food products assessment models. Similarly, effect of temperature on NIR spectra in solid wood samples of Norway spruce or OSB boards was studied by References [40,41]. They observed that with temperature increases, the two main hydroxyl absorbance bands at approximately 1450 and 1930 nm shifted by about 0.4 nm·°C^−1^ towards shorter wavelengths. More pronounced shift was noticed for heartwood due to higher moisture content in the sapwood. Ice crystals on wooden surfaces cause light scattering when sample temperature is below 0 °C. Consequently, it was recommended that the temperature should be taken into account in order to correctly predict moisture content, especially if it fluctuates around 0 °C [42]. The practical solution to compensate effect of temperature variation on the detector, measured object or ambience is a frequent measurement of dark and white references. This routine was also implemented in the presented research.

Work reported here was an essential part of the research project related to practical implementation of NIR spectroscopy for automatic grading of logs directly in the forest [43]. Extensive screening tests presented here resulted in development of a prototype hyperspectral imaging solution containing an array of sixteen micro-spectrometers simultaneously acquiring spatially resolved spectra from the surface of wood samples. All the instruments presented were also tested during real operation conditions while harvesting trees directly in the forest.

All the data used to support the findings of this study are available from the corresponding author upon request. The software codes used to program Arduino controllers are included as Appendix A and Appendix B.

## Figures and Tables

**Figure 1 sensors-20-00545-f001:**
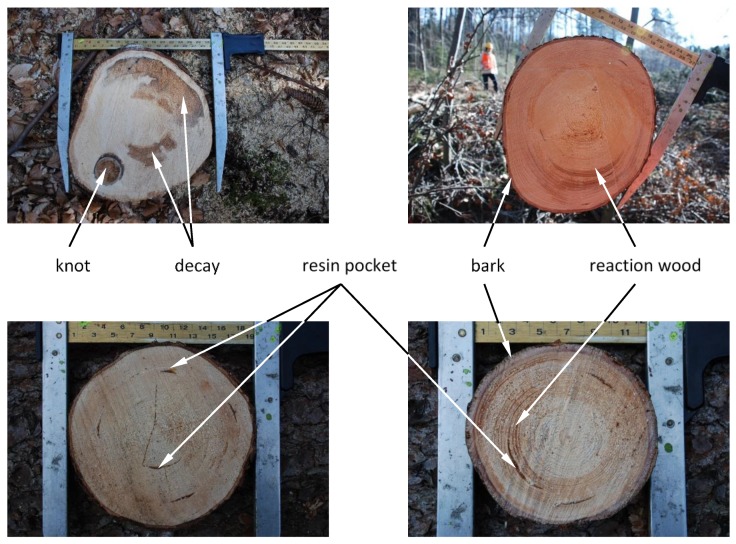
Experimental samples in a form of wooden disks containing diverse wood defects as used for spectroscopic measurements.

**Figure 2 sensors-20-00545-f002:**
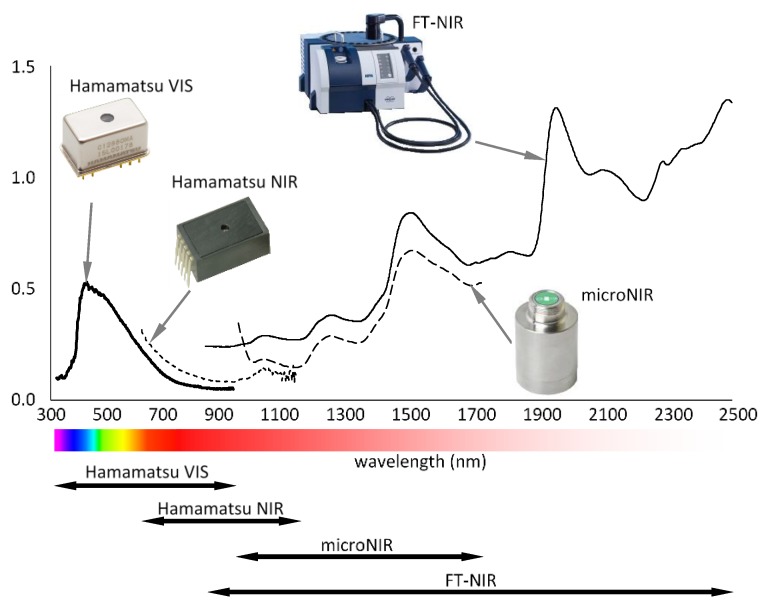
Absorbance spectra measured on the wood surface by different instruments evaluated.

**Figure 3 sensors-20-00545-f003:**
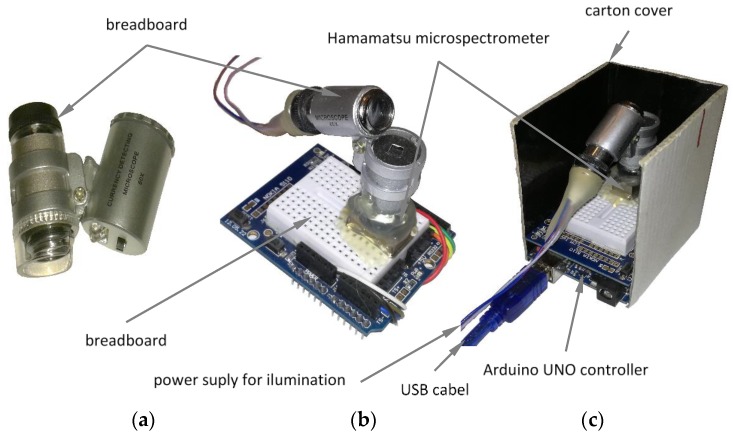
Focusing optics used in prototype spectrometers (**a**), the lenses, spectrometer with illumination installed on the breadboard (**b**) and assembled instruments ready for measurements (**c**).

**Figure 4 sensors-20-00545-f004:**
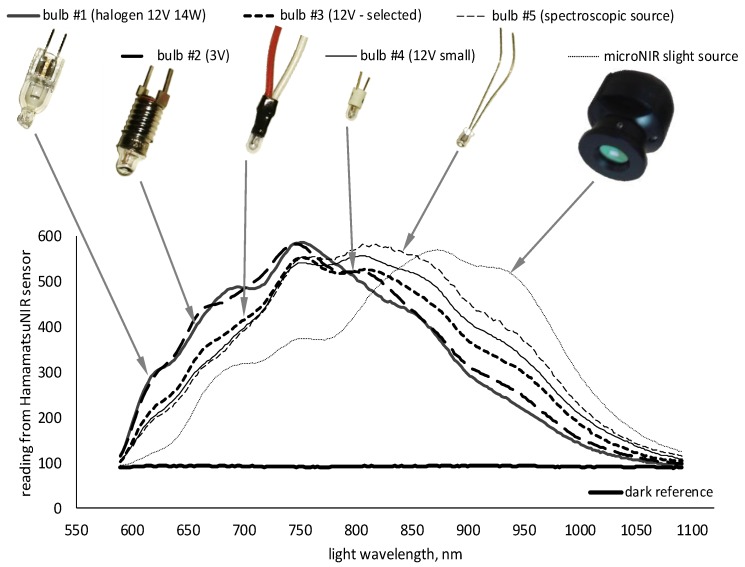
Different sources of light tested for suitability in developed prototypes and the spectral response recorded by the Hamamatsu NIR C11708MA sensor.

**Figure 5 sensors-20-00545-f005:**
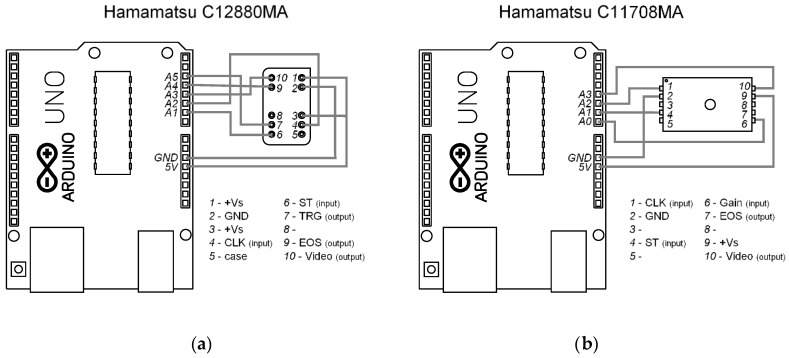
Connection diagram of Hamamatsu C12880MA (**a**) and C11708MA (**b**) micro-spectrometers to the Arduino UNO microcontroller.

**Figure 6 sensors-20-00545-f006:**
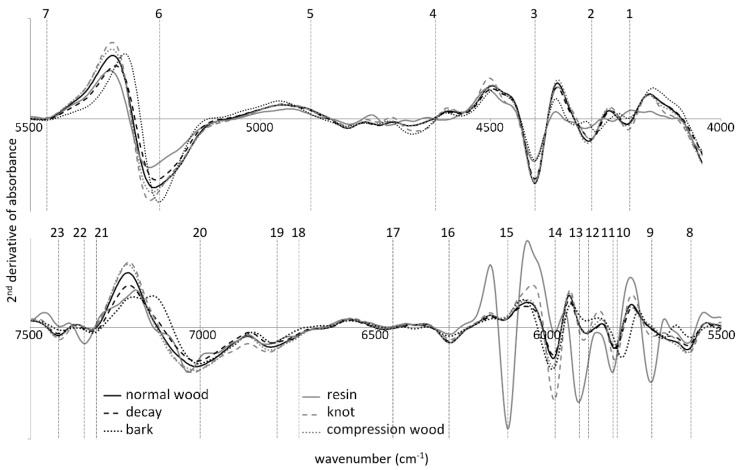
Model spectra of wood and wood defects detected by Fourier transform infrared (FT-NIR) instrument.

**Figure 7 sensors-20-00545-f007:**
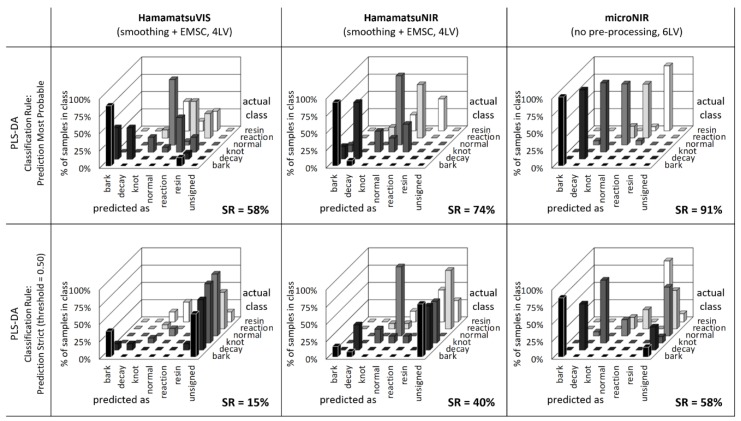
Automatic identification of defects on wood samples with Partial Least Squares discriminant analysis (PLSDA) (SR—success rate).

**Figure 8 sensors-20-00545-f008:**
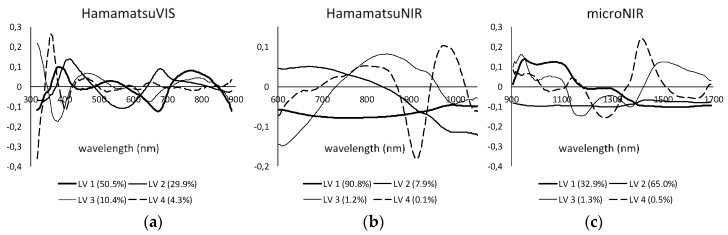
First four loadings (lateral variables LV) for PLSDA models of wood defects as obtained for HamamatsuVIS (**a**), HamamatsuNIR (**b**) and microNIR (**c**).

**Figure 9 sensors-20-00545-f009:**
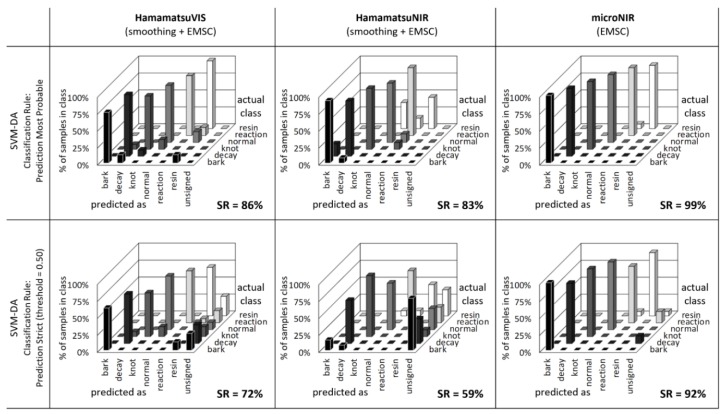
Automatic identification of defects on wood samples with Support Vector Machine (SVM) classification.

**Table 1 sensors-20-00545-t001:** Basic characteristic of tested spectrometers.

	Bruker MPA	MicroNIR Pro 1700	Hamamatsu NIR C11708MA	Hamamatsu VIS C12666MA
sensor technology	FT (Fourier Transform)	Linear Variable Filter	MEMS micro-electro-mechanical systems	MEMS micro-electro-mechanical systems
range (nm)	833–2500	950–1650	640–1050	340–780
resolution (nm)	0.8	6.2	20	15
weight (g)	15,000	64	9	5
portable	no	yes	yes	yes
instrument available on the market	yes	yes	no	no
measurement time (s) for a single spectrum	30	0.05–0.5	0.05–0.5	0.05–0.5

**Table 2 sensors-20-00545-t002:** Band assignment characteristic for wood, according to Schwanninger et al. [31] and Vagnini et al. [32].

	Wavenumber (cm^−1^)	Wavelength (nm)	Wood Component	Functional Group
1	4198	2382	holocellulose	CH
2	4280	2336	cellulose	CH, CH_2_
3	4404	2270	cellulose, hemicellulose	CH, CH_2_, OH, CO
4	4620	2164	cellulose, hemicellulose	OH, CH
5	4890	2044	cellulose semicrystalline and crystalline	OH, CH
6	5219	1916	water	OH
7	5464	1830	cellulose semicrystalline and crystalline	C=O
8	5587	1790	cellulose semicrystalline and crystalline	CH
9	5700	1754	extractives	CH_2_
10	5800	1724	hemicellulose (furanose/pyranose)	CH
11	5812	1720	extractives	CH_2_
12	5883	1700	hemicellulose	CH
13	5909	1692	extractives	CH
14	5980	1672	lignin	CH
15	6117	1635	extractives	CH_2_
16	6287	1590	cellulose crystalline	OH
17	6450	1550	cellulose crystalline	OH
18	6722	1487	cellulose semicrystalline	OH
19	6785	1474	cellulose	OH
20	7008	1426	amorphous cellulose/water	OH
21	7309	1368	aliphatic chains	CH
22	7344	1361	extractives	CH
23	7418	1348	aliphatic chains	CH

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
