# Peer review of "Development of Low-Cost Portable Spectrometers for Detection of Wood Defects"

_sensors, 2020, doi:10.3390/s20020545_

Round 1
Reviewer 1 Report
Abstract
Which surface of log? Transverse? Say log qualityFigure says cross section? How can one grade a log from cross section for knots etc.? This makes little sense.
Add cost of each system in text.
Table 2 should be before Fig 6 since the Fig. References the table.
Section 3.2 is alot of chemistry. Introduction needs good discusssion of chemistry. Introduction also lacks 2018 and 2019 papers. Here are 4 you need to add
Mechanical and physical properties of oriented strand board exposed to high temperature and humidity.... Potential use of Nir and vis spectroscopy to analyze chemical properties of thermally treated wood New pretreatment methods for vis-nir calibration modeling of air-dry density of.....2nd derivative adds higher noise to signal ratio. Discuss this.
What log grading rules was used?
Fig 2 does each system have a different scale like cm vs mm?
Discuss effect field temperature variance will have on NIR error.
Author Response
Dear Reviewer #1
Thank you very much for your time and valuable comments. We have revised our manuscript accordingly and we sincerely hope that it has been improved. Please find below detailed answers to your concerns, together with indication of the changes implemented in the manuscript.
Abstract
Which surface of log? Transverse? Say log quality
Figure says cross section? How can one grade a log from cross section for knots etc.? This makes little sense.
The spectroscopic system reported in our paper is a fruit of the broader project (SLOPE, as acknowledged in the latest chapter) that focused on the development of the intelligent processor head equipped with diverse sensors. Our overall goal was to provide a complete technological solution capable to automatically determine log quality at the earliest stage of the value chain. We identified measurement of cutting forces, free vibrations, stress wave velocity and VIS-NIR spectroscopy as most suitable solutions capable of objective assessment of wood properties. This can be a highly valuable information supporting operator to optimize cross cut location as well as to decide the quality grade/use of log. Each of the sensing method was useful to assess defined wood characteristics, however, none of these provided information directly corresponding to the established grading standards (EN 1309-3 for example). For that reason, we developed an alternative methodology where standardized quality classes (such as A, B, C or D) were replaced by “suitability indexes”. The detailed logic for this approach will be a separate report, to be published in a close future. In general, the information regarding presence of given defects is an input required by our expert system. The great challenge we faced when implementing SLOPE project was to identify optimal hardware configuration providing reliable information regarding log quality. The Manuscript we proposed to publish within frame of Sensors Journal is therefore a report from our studies where the most promising low-cost spectroscopic solutions were prototyped and tested in a pilot study.
We do agree with Reviewer comment regarding limitation of the spectroscopic assessment to scan only cross section of log. Indeed, the presence of knots may not be guaranteed and perhaps can be relatively low reliable information. However, proper identification and classification of knots is indispensable to deploy the automatic system we are proposing.
As the above explanation may clarify the general context of our study, we would like to propose to add the following text to the introduction chapter:
“The procedure for log quality grading is formalized and described in dedicated standards (e.g., EN 1309, EN 1316) or established as commonly recognized procedures [2]. Grading results in a quality class, typically A, B, C or D, with the first considered as superior. In that case, a trained expert visually detects all wood imperfections or defects as well as manually measures log dimensions and geometrical features. Commercial value of the resource is determined after combining all the available information into the unique quality grade. An important limitation of this approach is that it relies on visual assessment by a person, which, from its nature, is subjective and of limited reliability/repeatability. Industries, therefore, wish to adopt more objective methodologies for resource quality determination by integrating sensors and novel descriptors of the quality, better adjusted to specific needs of the value chain players [3].”
and to discussion chapter:
“The most convenient place to measure spectra with sensors developed in this research is on the cross section of logs after tree felling or on the pile of stored logs. Utilization of spectrometers to asses log sides is problematic as the bark layer prevents light from accessing the log’s interior. The only information available to scrutinize are traces of knots appearing on the side of the log visible over the bark layer. Even if determination of the log’s quality by assessing its cross section may be considered as limited, it can provide an enormous possibility to improve the state-of-the-art in quality grading and managing of value chains. The functionality of the proposed low-cost spectroscopic system can be extended to predict chemical composition of wood polymers as well as its distribution (map) over the log section or whole volume. It can lead toward development of new methodologies where raw resources are optimally destinated to down-stream conversion following the paradigm of “proper wood resources for proper use”.”
Add cost of each system in text.
The detailed cost of the system is a matter of individual “negotiations” with sensor provides, however following the Reviewer suggestion we included the following statement:
“Total cost of the hardware required for implementing the prototype in-field systems presented here varied from a few hundred (Hamamatsu sensors) to several thousands (MicroNIR sensor). However, additional resources should be considered when integrating these with a dedicated controller (instead of a laptop PC as used in this study) as well as all-weather cover and protection. Even then, overall investment to that solution is far less costly when compared to the corresponding laboratory equipment (FT-NIR) and still provides a unique set of quality characteristics impossible to acquire with alternative technologies.”
Table 2 should be before Fig 6 since the Fig. References the table.
Figure and table are re-located according to the suggestion.
Section 3.2 is alot of chemistry. Introduction needs good discusssion of chemistry. Introduction also lacks 2018 and 2019 papers. Here are 4 you need to add
Mechanical and physical properties of oriented strand board exposed to high temperature and humidity.... Potential use of Nir and vis spectroscopy to analyze chemical properties of thermally treated woodNew pretreatment methods for vis-nir calibration modeling of air-dry density of.....
The proper selection of the representing literature references is always a great challenge to balance depth of details for presenting research background and length of the paper. We did our very best in the original submission to include some of the relevant contributions. Therefore, we cited 14 papers in introduction that were related to the application of NIR in the context of this research. By considering Reviewer’s suggestion, we would like to include additional papers that may help reader to better follow our research:
Li, B.K. Via, Q. Cheng, Y. Li, “Lifting Wavelet Transform De-noising for Model Optimization of Vis-NIR Spectroscopy to Predict Wood Tracheid Length in Trees”, Sensors, 18, 4306, 2018. Schimleck, F. Antony, C. Mora, J. Dahlen, “Comparison of Whole-Tree Wood Property Maps for 13- and 22-Year-Old Loblolly Pine”, Forests, 9, 287, 2018. Schimleck, J.L.M. Matos, R. Trianoski, J.G. Prata, “Comparison of methods for estimating mechanical properties of wood by NIR spectroscopy”, J. Spectrosc. 1–10, 2018We have also carefully considered other papers suggested by the Reviewer. All are highly interesting contributions, and after internal discussions we would like to propose to add reference paper #2 and #3 to the introductory chapter. The paper #3 is referenced in discussion when effect of temperature on the spectra is included following Reviewer suggestion as described below.
2nd derivative adds higher noise to signal ratio. Discuss this.
The optimal setting of the NIR spectra pre-processing is the most challenging part of any chemometric analysis. We are fully aware of the important limitations of the second derivative and tested several alternative algorithms when preparing our models (as described in detail in Material and Methods chapter). However, second derivate resulted as the best solutions and therefore we limited the description in the report. Following Reviewer suggestion, we would like to include the following statement in the manuscript text, together with additional literature reference:
“An unwanted effect of the derivatization process is that the signal-to-noise ratio decreases at higher orders of derivatives. It is a consequence of the discrimination effect and the fact that, by its nature, the noise contains the sharpest features in the spectrum. As a consequence, the bandwidth of noise corresponds to the interval of the spectral data used in the derivative calculation. The decrease in signal-to-noise ratio can be reduced by using smoothing properties of the Savitzky-Golay algorithm, even if a high degree of smoothing can distort the derivative spectrum [33].”
A.J. Owen, “Uses of Derivative Spectroscopy”, Application Note: UV-Visible Spectroscopy, Agilent Technologies, pp. 8
What log grading rules was used?
As already described above, the reported research is a part of the multisensory system described in the paper we are refereeing in our Manuscript:
Jakub Sandak, Anna Sandak, Stefano Marrazza, Gianni Picchi (2019) Development of a sensorized timber processor head. Croatian Journal of Forest Engineering 40(1): 25-37
Our reference grading rules were according to work of Pollini and more general EN 1309-3.
Pollini C. Manuale per la classificazione visuale qualitativa del legno tondo di Abete Rosso, Abete Bianco e Larice 2006
EN 1309-3 Round and sawn timber - Methods of measurements - Part 3: Features and biological degradations, STANDARD by DIN-adopted European Standard, 09/01/2018
However, the ultimate grading algorithm for our solution bases on the suitability index instead of quality class. It was not reported yet in any report/publication and we are intensively working on dedicated manuscript. However, following Reviewer’s comment, we included the additional text and two references. These are part of the introduction chapter and are already presented in this letter (response to the first concern of Reviewer #1).
Fig 2 does each system have a different scale like cm vs mm?
There is a scale in Figure 2, but after reading Reviewer’s comment, we realized that perhaps it was not enough apparent. We have modified Figure 2 to simplify understanding and allow more intuitive interpretation.
Discuss effect field temperature variance will have on NIR error.
We do agree with the concern of Reviewer and are aware of the temperature effect on the measured spectra/derived models. It is a very complex phenomenon, as there are several disturbance sources affecting performance of spectroscopic (opto-electronic) system. In general, frequent measurement of the black and white reference is a practical solution to control variability of data due to temperature variations. That was an approach that we have implemented in our measurement routine. However, it was not explicitly defined in the original version of the manuscript that has been augmented in the revised version:
“Previous studies [38] and [39] reported that temperature variations degrade the quality of agri-food products assessment models. Similarly, effect of temperature on NIR spectra in solid wood samples of Norway spruce or OSB boards was studied by [40] and [41]. They observed that with temperature increases, the two main hydroxyl absorbance bands at approximately 1450 and 1930 nm shifted by about 0.4 nm·°C−1 towards shorter wavelengths. More pronounced shift was noticed for heartwood due to higher moisture content in the sapwood. Ice crystals on wooden surfaces cause light scattering when sample temperature is below 0°C. Consequently, it was recommended that the temperature should be taken into account in order to correctly predict moisture content, especially if it fluctuates around 0°C [42]. The practical solution to compensate effect of temperature variation on the detector, measured object or ambience is a frequent measurement of dark and white references. This routine was also implemented in the presented research.”
Summarizing, we would like to thank you again for your valuable comments. We have revised the manuscript accordingly, considering also Reviewer #2 suggestions. We do hope that our Manuscript is improved and may be re-considered for publication in the Sensors Journal.
With sincerely regards,
Jakub Sandak, on behalf of authors

Reviewer 2 Report
The authors modified and designed three portable spectral detectors covering the visible near infrared region., implements the detailed detection of defects natural wood inside, this work compared with the traditional natural defect detection method and the equipment has some optimization and improvement, but there are some problems as well as the information is not completely account, I believe that publication of the manuscript may be considered only after the following issues have been resolved.
The font of the information such as the legend in the graph of the article is too small, which affects normal reading. I hope the author can check and modify the format. The author gives the code used in the spectral detector in Appendix A of the article, but it is obviously difficult for ordinary people to understand what it means. Therefore, I hope that the author can provide a flowchart matching the code in the appendix and illustrate it with the model in the article. At the end of the article, the author mentioned “Partial Least Squares discriminant analysis” and “Automatic identification of defects on wood samples with Support Vector Machine classification”, but they were not explained in detail. In addition, the meanings specifically represented in Figures 7 and 9 have not been explained clearly in the article, and I hope the authors can combine explanations. The portable spectrometers designed in this paper are all modifications of other companies' products, while the author only miniaturizes the equipment through the work of separation and combination. I hope the author can show some innovation in this article.
Author Response
Dear Reviewer #2
Thank you very much for your time and valuable comments. We have revised our manuscript accordingly and we sincerely hope that it has been improved. Please find below detailed answers to your concerns, together with indication of the changes implemented in the manuscript.
The authors modified and designed three portable spectral detectors covering the visible near infrared region., implements the detailed detection of defects natural wood inside, this work compared with the traditional natural defect detection method and the equipment has some optimization and improvement, but there are some problems as well as the information is not completely account, I believe that publication of the manuscript may be considered only after the following issues have been resolved.
Thank you very much for your kind evaluation. We did all our best to modify the manuscript according to your and Reviewer #1 suggestions. We do hope that we were able to properly identify all the issues you raised.
The font of the information such as the legend in the graph of the article is too small, which affects normal reading. I hope the author can check and modify the format.
All figures were revised and increased size of fonts to simplify reading, as suggested.
The author gives the code used in the spectral detector in Appendix A of the article, but it is obviously difficult for ordinary people to understand what it means. Therefore, I hope that the author can provide a flowchart matching the code in the appendix and illustrate it with the model in the article.
We do believe that providing a source code of our software as an Appendix can be useful for all Readers willing to replicate our solution. The code is a direct implementation of the sensor’s interface protocol as defined by the producer (Hamamatsu) optimized for the low-cost open-hardware controller (Arduino). There is not any specific flowchart related to the software code to be provided as an addition to the code. However, following suggestion of the Reviewer, we have provided/highlighted additional comments in the code as well as specific references to the timing chart of each sensor defined as a part of the official documentation of the product. Unfortunately, we do not have a copy right to include the figure bellow in our Manuscript, therefore, we are providing links to the Websites fully describing the functionality of both sensors. Both timing charts are included in the revised version of both Appendixes, but again, we are not sure if we are authorized to include the use of materials from official user manuals provided by sensor manufacturers.
Moreover, we would suggest to Editor to let us prepare Appendixes as a separate files to be downloaded from the repository server by any Reader interested in replication of our solution.
We sincerely believe that each Reader with a basic (entrance) knowledge on the Arduino technology will be able to properly implement the code and set/use the sensor in her/his application on the basis of the information provided within the frame of our Manuscript and technical data sheets published by producer.
At the end of the article, the author mentioned “Partial Least Squares discriminant analysis” and “Automatic identification of defects on wood samples with Support Vector Machine classification”, but they were not explained in detail.
Both methods used in our research are relatively well described in the literature and are usual approaches for chemometric analysis of spectroscopic data. However, following recommendation of Reviewer #2, the following explanation is included in the revised version of the Manuscript, together with literature reference:
“PLS-DA is widely used in the analysis of multivariate data. It helps to determine if groups of samples are distinct and identify all the spectral features that can describe the differences between groups. All these differences are expressed as model loadings or latent variables. An important advantage of PLS-DA is its availability to handle highly colinear and noisy data. Moreover, it provides a visual interpretation of a complex dataset through easily interpretable scores plots that highlight the differences between groups.”
and
“Support Vector Machine is a supervised machine learning algorithm useful for solving both classification or regression problems. In this approach, data are presented in n-dimensional space, corresponding to the number of variables describing the data. In contrast to PLS-DA, SVM is not influenced by the distribution of diverse sample classes. The methods allow flexibility in the choice of kernel function that leads to the separation of two groups of samples by solving either linear or non-linear problems. An important limitation is that SVM does not provide interpretable model’s statistics, such as scores or loadings [37].”
In addition, the meanings specifically represented in Figures 7 and 9 have not been explained clearly in the article, and I hope the authors can combine explanations.
We did our best to better explain meaning and interpretation of results presented in Figure 7 and 9 in the revised manuscript. The size of fonts has been increased as well as following text included:
“All columns located on the diagonal running from bottom left to top right (excluding unsigned row) correspond to the properly identified samples. In that case, the spectra of a given defect was identified as a member of the class describing exactly such defect. Therefore, it is desired that the optimal discrimination model results in majority of validation samples appearing on the diagonal of the confusion table. All the other spectra (not laying on the table diagonal) are considered as undesirable or wrong classifications. The case when normal wood is identified as defect is defined as false negative, in contrast to false positive when normal wood as identified as deficient. A convenient numerical quantifier of the discrimination model accuracy is the success rate (SR) computed as a percent ratio of the correctly classified to total number of samples.”
The portable spectrometers designed in this paper are all modifications of other companies' products, while the author only miniaturizes the equipment through the work of separation and combination. I hope the author can show some innovation in this article.
If we correctly understand, the impression of Reviewer #2 is a low innovation of our approach reported in this paper. We are aware that some part of our work can be considered as a simple integration of available technical solutions, especially when regarding the sensing elements (spectrometers). However, the most important innovation is a possibility for using (for a first time) low-cost spectroscopic sensors in such a particular in-field application, as assessment of the log/timber properties at the harvesting site. Our ambition was to develop a sound solution that can be easily replicated and adopted for general forest/wood value chain applications. It is a field of a great market/research potential, where spectroscopy can revolutionize state-of-the-art practices. The follow-up of this research (still ongoing project) is further extension of the presented hardware solution by integrating several spectrometers in a form of a sensor network array. This may provide a unique functionality corresponding to hyperspectral imaging, still assuring high resistance to harsh environmental conditions present in forest operations. All the experiences, hardware selections, technical solutions as well as data analysis algorithms developed in this research are fundamental for the proper implementation and therefore this Manuscript can support other researches in further developments in diverse application fields.
Summarizing, we would like to thank you again for your valuable comments. We have revised the manuscript accordingly, considering also Reviewer #1 suggestions. We do hope that our Manuscript is improved and may be re-considered for publication in the Sensors Journal.
With sincerely regards,
Jakub Sandak, on behalf of authors

Round 2
Reviewer 1 Report
The authors did an excellent job responding to the reviewers